# Response of Soil Organic Carbon Stock to Bryophyte Removal Is Regulated by Forest Types in Southwest China

Deyun Chen [1,2], Mutian Cai [1,2], Debao Li [1,2], Shiming Yang [1,2] and Jianping Wu [1,2,*]

1. Yunnan Key Laboratory of Plant Reproductive Adaptation and Evolutionary Ecology and Centre for Invasion Biology, Institute of Biodiversity, School of Ecology and Environmental Science, Yunnan University, Kunming 650500, China
2. Key Laboratory of Soil Ecology and Health in Universities of Yunnan Province, School of Ecology and Environmental Science, Yunnan University, Kunming 650500, China
* Correspondence: jianping.wu@yun.edu.cn

**Abstract:** Bryophytes play an important role in biogeochemical cycles and functions in forest ecosystems. Global climate changes have led to the population decline of bryophytes; however, the effects of bryophyte loss on the soil organic carbon stock and microbial dynamic remain poorly understood. Here, bryophytes were artificially removed to simulate the loss of bryophytes in two forests in Southwest China, i.e., evergreen broad-leaved forest and temperate coniferous forest. Soil physicochemical properties, microorganisms, and soil organic carbon stocks were analyzed and factors regulating soil organic carbon stocks were explored. Results showed that bryophyte removal significantly decreased soil organic carbon in the coniferous forest but had a negligible effect on the evergreen broad-leaved forest. Bryophyte removal had an insignificant effect on soil properties and microbial PLFAs except that soil nitrogen significantly increased in the 0–10 cm layer in the evergreen broad-leaved forest, while soil temperature and bulk density increased in the coniferous forest in the 0–10 and 10–20 soil layers, respectively. Soil organic carbon stocks increased by 14.06% in the evergreen forest and decreased by 14.39% in the coniferous forest. In the evergreen forest, most soil properties and microorganisms contributed to the change of soil organic carbon stocks, however, only soil organic carbon and depth had significant effects in the coniferous forest. Our findings suggest that soil physiochemical properties and microorganisms regulated the different responses of soil organic carbon stocks after bryophyte removal in the two forests. More research is needed to better understand the effects of understory plants on soil organic carbon stocks in various forest ecosystems.

**Keywords:** evergreen broad-leaved forest; temperate coniferous forest; bryophyte removal; soil organic carbon stock; understory plant; PLFAs



## 1. Introduction

Global climate change is one of the most significant challenges humans are facing in the 21st century [1,2]. Climate change is a serious ecological issue reducing sustainable human development and affecting the balance of global ecosystems [1,3]. As global climate change intensifies, the future uncertainty of climate change makes it difficult to predict the response of the terrestrial carbon (C) cycle [4], especially for forest ecosystems [5]. Forest ecosystems are an important carbon pool. The fixed C in forest vegetation is approximately 75% of total terrestrial vegetation [6]. Forest soil C store also accounts for nearly half of the total carbon stored in terrestrial soil [7]. Thus, small changes in the forest's soil organic carbon (SOC) pool may have a significant impact on the global climate [8]. Additionally, global-scale climate changes have the capacity to influence plant photosynthesis and productivity and can lead to loss of plant diversity [9]. Climate change disturbs the balance between above- and below-ground communities, directly and indirectly affecting C turnover in forest ecosystems [10].

Understory plants have an important impact on soil C and nitrogen (N) cycle by affecting microorganisms [11]. For example, the removal of understory plants decreased litter decomposition and affected microbial community composition [12], increased soil surface temperature, and the rate of N mineralization [13]. Other studies showed understory removal reduced fine root biomass, SOC content, and mineralization of nitrogen [14,15]. Bryophytes typically serve as an indicator species of ecosystem productivity. Decreases in bryophyte abundance may have a negative impact on ecosystem productivity [16,17]. The intensification and phenomenon of climate and land use change is threatening bryophyte species composition, distribution, and richness [18]. The life stages of bryophytes are sensitive to the rise of surface soil temperature, the change of precipitation pattern [19], the increased N deposition [20], the enhanced radiation [21], and the increase of $CO_2$ concentration [22]. Bryophytes are important primary producers in forests, for example, the net primary productivity (NPP) of bryophytes accounts for 20–50% of ecosystem NPP and the C accumulated by bryophyte photosynthesis accounts for approximately 5% of the total primary production of trees in a boreal forest [23]. Another study showed that *Sphagnum junghuhnianum* accounted for 50% of the entire black spruce forest photosynthesis in summer [24]. Bryophytes can grow an abundant number of polyphenols and peat alcohols, which are difficult to decompose, resulting in the accumulation of biomass residue and strong C-fixing ability [25]. In addition, bryophytes are not only important producers with a simple structure and wide distribution [26], but also maintain forest ecosystem stability [27,28]. However, the ecological functions of bryophytes have not been fully understood [29] and how the loss of bryophyte affects SOC in various forest ecosystems remains unclear.

As an active part of soil, SOC plays an important role in soil productivity and the carbon cycle [30]. The dynamics of SOC are primarily affected by climate [10], soil properties [31], human activities [32], and vegetation cover [33]. The status of the soil microbial community is considered an indicator of the soil ecosystem [34] and plays an important role in regulating soil carbon dynamics and plant growth [35,36]. Soil microorganisms also affect the process of soil nutrient cycling and various geochemical cycles [37]. Understory plant removal affects SOC by decreasing the diversity and function of soil microorganisms [12,38,39]. For example, bryophytes are considered an important driving force in global biogeochemical cycles in terrestrial ecosystems [40,41]. Bryophytes have direct and indirect effects on C cycles through photosynthesis, respiration, and secondary metabolism [42]. Yet, the responses of soil microorganisms and spatial variation of ecosystem types to ground bryophyte removal have not been well explored.

In this study, a subtropical humid evergreen broad-leaved forest in Ailaoshan of Yunnan Province (Ailaoshan) and a cold temperate coniferous forest in Bitahai of Yunnan Province (Bitahai) were selected in Southwest China. We investigated the effects of bryophyte removal on soil physicochemical properties and microorganisms by simultaneously removing bryophytes in both subtropical and cold temperate forests. Since bryophytes can increase soil water content [43] and reduce surface temperature [25], which in turn affects the decomposition of organic matter and geochemical cycles. We first hypothesized that bryophyte removal would reduce SOC and thus soil organic carbon stock in both forests. The Ailaoshan is an old-growth forest with high temperature and abundant precipitation and Bitahai has relatively low temperature and less precipitation. Thus, we also hypothesized that soil microorganisms in subtropical evergreen broad-leaved forests would be less responsive to bryophyte removal than in cold temperate coniferous forests.

## 2. Materials and Methods

### 2.1. Study Area

The study was simultaneously conducted in the subtropical and cold temperate forests in Yunnan Province, China. For subtropical forest, the sampling site was located in Ailaoshan Station for Subtropical Forest Ecosystem Studies (latitude and longitude: 24°32′ N, 101°01′ E; altitude: 2450 m, Figure 1). The mean annual temperature is 11.3 °C

and the mean annual precipitation is 1840 mm yr$^{-1}$. The climate type is subtropical monsoon climate. The mean soil organic carbon content was 116 g kg$^{-1}$, the mean soil total nitrogen content was 7 g kg$^{-1}$, and the mean pH value was 4.2. The dominant tree species included *Lithocarpus crassifolius*, *Clethra delavayi,* and *Rhododendron irroratum*, and the dominant bryophyte species was *Sphagnum junhuhnianum* [44,45]. For the cold temperate forest, the sampling site was located in Bitahai Nature Reserve (latitude and longitude: 27°46′ N, 108°32′ E; altitude: 3500 m, Figure 1). The mean annual temperature is 5.9 °C and the mean annual precipitation is 618 mm yr$^{-1}$. The climate type is a cold temperate monsoon climate. The mean soil organic carbon content was 93.54 g kg$^{-1}$, the mean soil total nitrogen content was 5.23 g kg$^{-1}$, and the mean pH value was 4.7. The dominant tree species include *Picea asperata*, *Abies fabri*, and *Usnea longissima*, and the dominant bryophyte species is *Actinothuidium hookeri* [46,47].

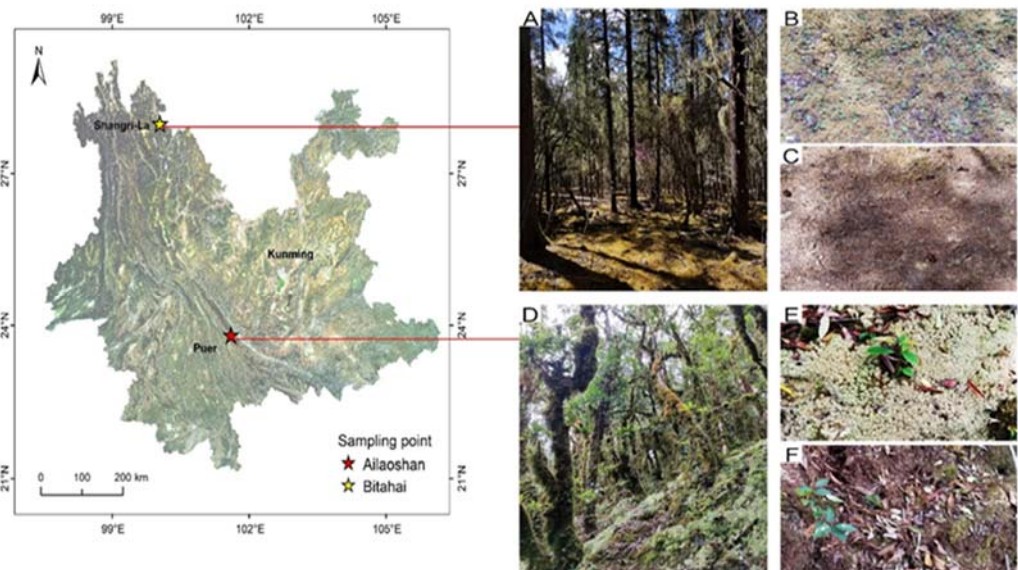

**Figure 1.** Schematic diagram of sampling sites and experimental treatments. Ailaoshan sampling site: Subtropical evergreen broad-leaved forest in Ailaoshan. Bitahai sampling site: Cold temperate coniferous forest in Bitahai. (**A**): Habitat at the experimental site in Ailaoshan, (**B**): No bryophyte removal in Ailaoshan, (**C**): Bryophyte removal in Ailaoshan, (**D**): Habitat at the experimental site in Bitahai, (**E**): No bryophyte removal in Bitahai, (**F**): bryohyte removal in Bitahai.

*2.2. Experimental Design and Sample Collection*

The study in Ailaoshan and Bitahai was set up in May 2017 according to the randomized block design. No bryophyte removal (CK) and live bryophyte removal (BR) with six repetitive blocks and twelve plots were established in each site. For bryophyte removal treatment, live and dead bryophytes on the soil surface were artificially removed. All plots were maintained monthly after the first removal of bryophytes. Because bryophytes in Ailaoshan were relatively patchy, the area of each block is set as 1 × 1 m$^2$. While the terrain is relatively flat in Bitahai, bryophytes are evenly and widely distributed, and the quadrat size was set as 3 × 3 m$^2$. Each block is separated by more than 3 m to avoid mutual interference. In May 2018, three soil cores (3 cm diameter) were collected from 0–10 cm and 10–20 cm in each plot. The soil of three cores was mixed to form one composited sample per plot. A total of 48 composited samples were collected in the two forests. Plant litter and stones were removed from the soil. In addition, a 100 cm$^3$ stainless steel cylinder (diameter = 5 cm, height = 5 cm) was used to collect soil bulk density in each soil layer of each plot. Soil samples were divided into two parts, one part (air dry) was used for the determination of soil physicochemical properties and the other part (fresh soil) was stored at −80 °C to analyze PLFAs.

### 2.3. Soil Measurements

Soil bulk density (SBD) was measured using the drying method. Soil water content (SWC) (g of water/100 g dry soil) was examined gravimetrically by drying fresh soil at 105 °C to constant weight. Soil temperature (ST) at 10 cm depth was measured by a soil temperature probe, and it was repeated three times for 30 seconds each time. Soil pH was determined using a 1:2.5 (wt/vol) ratio of soil to deionized water. SOC and dissolved organic carbon (DOC) were determined using a merged Vario TOC analyzer (Vario TOC, Langenselbold, Germany), SOC was determined by dry combustion at 980 °C in solid mode, DOC was extracted with 0.05 mol L$^{-1}$ K$_2$SO$_4$ for 30 min and was determined in liquid mode. Soil total nitrogen (TN) concentration was determined by the sodium salicylate method and using an auto discrete analyzer (De Chem-Tech. GmbH, CleverChem380, Hamburg Germany) [47,48]. Soil C to N ratio (C:N) was calculated as the ratio of SOC to TN. Soil organic carbon stocks (SOCs) was calculated using the following formula [49]:

$$SOCs = SOC \times SBD \times SD/10 \qquad (1)$$

SOCs is the SOC stock (mg ha$^{-1}$), SOC is the soil organic carbon content (g kg$^{-1}$), SBD is the soil bulk density (g cm$^{-3}$), and SD is the soil layer thickness (cm).

The soil microbial community was characterized using phospholipid fatty acids (PLFAs) analysis as described by Bossio & Scow [50]. Briefly, 8 g fresh soil was weighted, then PLFAs were extracted from the soil with a single-phase mixture of chloroform-methanol-citrate buffer (1:2:0.8, v/v/v; 0.15 mol, pH 4.0). After extraction, the lipids were divided into neutral lipids, glycolipids, and polar lipids (phospholipids) on a silicic acid column. The phospholipids were methylated and separated on a gas chromatograph equipped with a flame ionization detector. Peak areas were quantified by adding methyl nonadecanoate fatty acid (19:0) as the internal standard before the methylation step. The resulting fatty acid methyl esters were then separated and identified by gas chromatography (Agilent 6890N, Wilmington, DE, USA) fitted with a MIDI Sherlock® microbial identification system (MIDI, Inc., Newark, NJ, USA). For each sample, different PLFAs represented different groups of soil microorganisms. The abundance of individual fatty acids was determined as nmol per g of dry soil. Bacterial PLFAs were represented by 15:0 iso, 15:0 anteiso, 15:0, 16:0 iso, 16:1ω9, 17:0 iso, 17:0 anteiso, 17:1ω8c, 17:0, 17:0 cyclo, 18:1ω7c, and 19:0 cyclo. Fungi were represented by the PLFAs 18:1ω9 and 18:2ω6. The ratio of 18:2 × 6 to total bacterial PLFAs was used to estimate the ratio of fungal to bacterial biomass (F:B) in soils. Gram-positive bacteria were considered to be represented by PLFAs i15:0, a15:0, i16:0, i17:0, and a17:0, whereas gram-negative bacteria were considered to be represented by PLFAs 16:1ω7, cy17:0, cy19:0, and 18:1ω7. Taken together, all of the PLFAs indicated above were considered to be representative of the total PLFAs of the soil microbial community [12]. Alpha diversity includes the Chao1 index (abundance of PLFA's types) and Shannon index (diversity and evenness of PLFA's types), which were calculated using PLFA's types. The first axis from the principal coordinate analysis of Bray–Curtis distance was used as the microbial community beta diversity.

### 2.4. Statistical Analyses

All statistical analyses were performed with R version 3.6.3 [51]. A Kruskal–Wallis test was used to determine the effect of bryophyte removal and soil depth on soil properties, soil microbial PLFAs, and alpha diversity index. Dunn's method was used for multiple comparisons. All data were standardized and then subjected to multivariate analysis of variance (ANOVA). The effects of bryophyte removal, soil depth, and interactions on soil physicochemical properties and soil microorganisms were analyzed by two-way ANOVA. The effects of bryophyte removal, soil depth, forest type, and interactions on soil physicochemical properties and soil microorganisms were analyzed by three-way ANOVA. Second, redundancy (RDA) analysis was used to study the relationships between soil

microbial community composition and physicochemical properties. Random forest model was used to predict the important factors on SOCs at the two sites.

## 3. Results

### 3.1. Responses of Soil Physicochemical Properties and PLFAs

Bryophyte removal rarely had a significant effect on soil physicochemical properties and microbial PLFAs in both forest types (Figures 2, 3 and A1). In the evergreen forest, bryophyte removal significantly increased SBD (Figure 2F) and ST (Figure 2I) in the 0–10 cm layer. Two-way ANOVA showed bryophyte removal significantly increased SOCs. Soil depth significantly decreased SOC, SOCs, TN, SWC, total PLFAs, fungal PLFAs, and bacterial PLFAs, and significantly increased pH and SBD (Table 1). In the coniferous forest, bryophyte removal significantly increased TN (Figure 2C) in the 0–10 cm layer and significantly decreased the soil C:N ratio in the 0–10 cm layer (Figure 2D). Two-way ANOVA showed bryophyte removal significantly increased SOC. Soil depth significantly decreased SOC, SOCs, TN, total PLFAs, bacterial PLFAs, and fungal PLFAs, and significantly increased pH (Table 1). Three-way ANOVAs showed forest type had significant effects on SOC, SBD, SOCs, pH, SWC, total PLFAs, bacterial PLFAs, and fungal PLFAs (Table A1). Bryophyte removal significantly changed SOC and SBD. Soil TN was significantly influenced by the interaction of bryophyte removal, soil depth, and forest type (Table A1).

**Table 1.** Bryophyte removal and soil depth were used for two-way ANOVA in the two forests.

| | | Ailaoshan | | | Bitahai | | |
|---|---|---|---|---|---|---|---|
| | | **BR** | **SD** | **BR × SD** | **BR** | **SD** | **BR × SD** |
| SOC | *F* | 0.38 | 87.87 | 0.44 | 4.62 | 50.40 | 0.71 |
| | *P* | 0.55 | <0.001 | 0.51 | 0.04 | <0.001 | 0.41 |
| DOC | *F* | 2.42 | 2.67 | 1.38 | 0.41 | 1.46 | 0.22 |
| | *P* | 0.14 | 0.12 | 0.26 | 0.53 | 0.24 | 0.65 |
| SOCs | *F* | 4.19 | 61.20 | 1.20 | 2.42 | 33.88 | 0.21 |
| | *P* | 0.05 | <0.001 | 0.29 | 0.14 | <0.001 | 0.65 |
| TN | *F* | 0.20 | 53.91 | 0.13 | 3.31 | 7.24 | 11.94 |
| | *P* | 0.66 | <0.001 | 0.73 | 0.08 | 0.01 | 0.003 |
| pH | *F* | 011 | 57.28 | 0.91 | 0.62 | 7.34 | 1.40 |
| | *P* | 0.75 | <0.001 | 0.35 | 0.44 | 0.01 | 0.25 |
| SWC | *F* | 1.20 | 5.56 | 0.25 | 0.33 | 0.07 | 0.05 |
| | *P* | 0.29 | 0.03 | 0.62 | 0.57 | 0.79 | 0.83 |
| SBD | *F* | 7.57 | 12.91 | 0.71 | 1.17 | 2.44 | 0.10 |
| | *P* | 0.01 | 0.002 | 0.41 | 0.29 | 0.13 | 0.76 |
| T PLFAs | *F* | 1.21 | 41.98 | 0.12 | 0.19 | 8.24 | 0.10 |
| | *P* | 0.28 | <0.001 | 0.73 | 0.67 | 0.01 | 0.76 |
| B PLFAs | *F* | 0.04 | 41.75 | 0.13 | 0.23 | 8.39 | 0.09 |
| | *P* | 0.84 | <0.001 | 0.72 | 0.64 | 0.01 | 0.77 |
| F PLFAs | *F* | 0.82 | 35.97 | 0.18 | 0.06 | 7.76 | 0.26 |
| | *P* | 0.37 | <0.001 | 0.68 | 0.81 | 0.01 | 0.62 |
| Chao1 | *F* | 0.49 | 12.35 | 2.13 | 2.64 | 0.21 | 0.34 |
| | *P* | 0.49 | 0.002 | 0.16 | 0.01 | 0.66 | 0.57 |
| Beta | *F* | 4.05 | 0.09 | 0.27 | 0.45 | 0.01 | 2.08 |
| | *P* | 0.06 | 0.77 | 0.61 | 0.51 | 0.94 | 0.16 |

Note: *F* and *P* values for the effects of bryophyte removal (BR), soil depth (SD) and interactions on soil physicochemical properties and soil microorganisms. SOC: soil organic carbon, DOC: soil dissolved organic carbon, TN: soil total nitrogen, SOCs: soil organic carbon stocks, SWC: soil water content, SBD: soil bulk density, Chao1: microbial Chao1 index, Beta: microbial beta diversity, T PLFAs: total microbial PLFAs, B PLFAs: bacterial PLFAs, F PLFAs: fungal PLFAs.

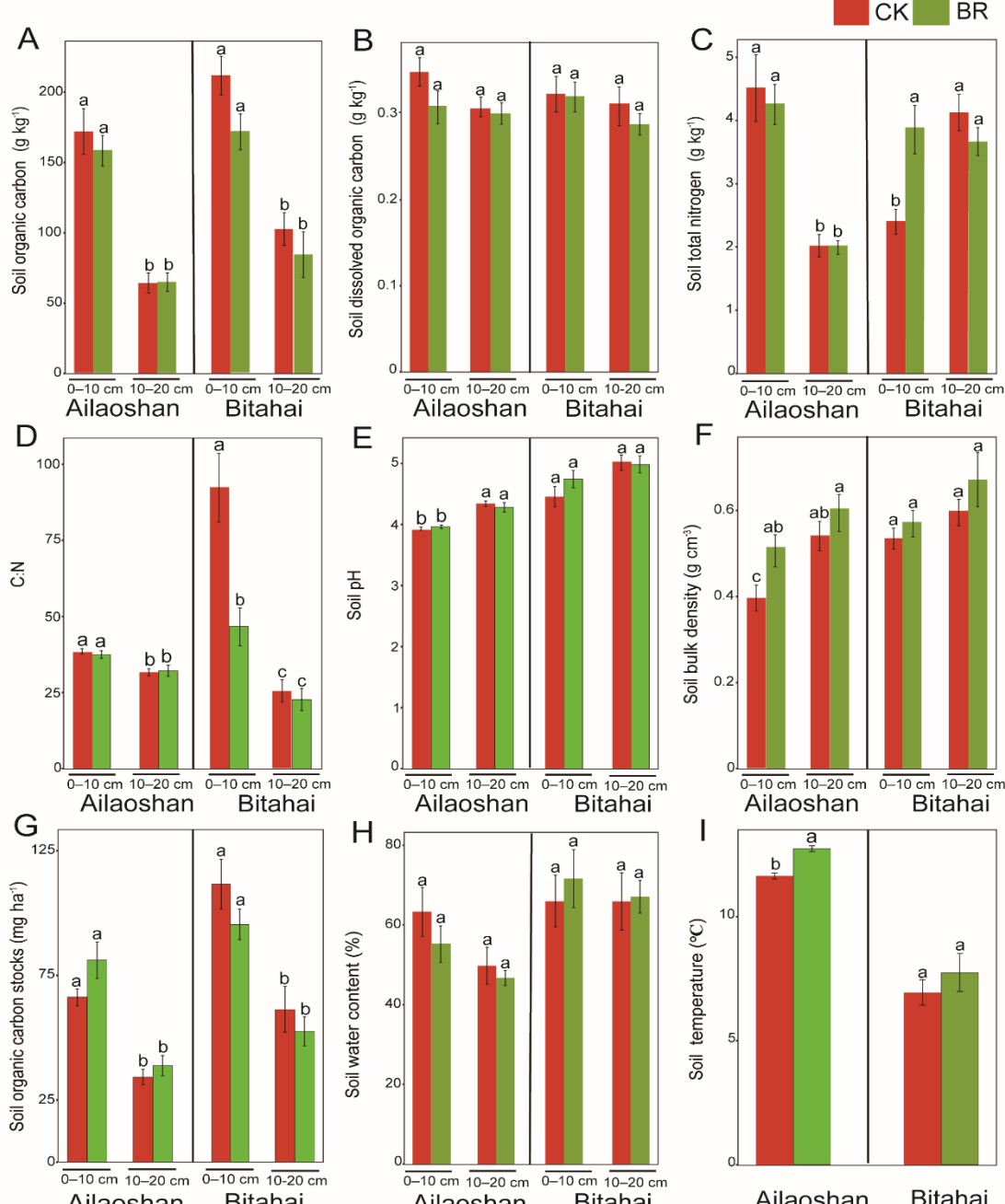

**Figure 2.** Effects of bryophyte removal (CK: no bryophyte removal; BR: (bryophytes removed) and soil depth (0–10 and 10–20 cm) on soil physicochemical properties in Ailaoshan and Bitahai. (**A**): soil organic carbon, (**B**): soil dissolved organic carbon, (**C**): soil total nitrogen, (**D**): soil organic carbon and total nitrogen, (**E**): soil pH, (**F**): soil bulk density, (**G**): soil organic carbon stocks, (**H**): soil water content, (**I**): soil temperature. Values are the means ± SE of six plots. Different lowercase letters mean significant differences at *P* < 0.05 between treatments.

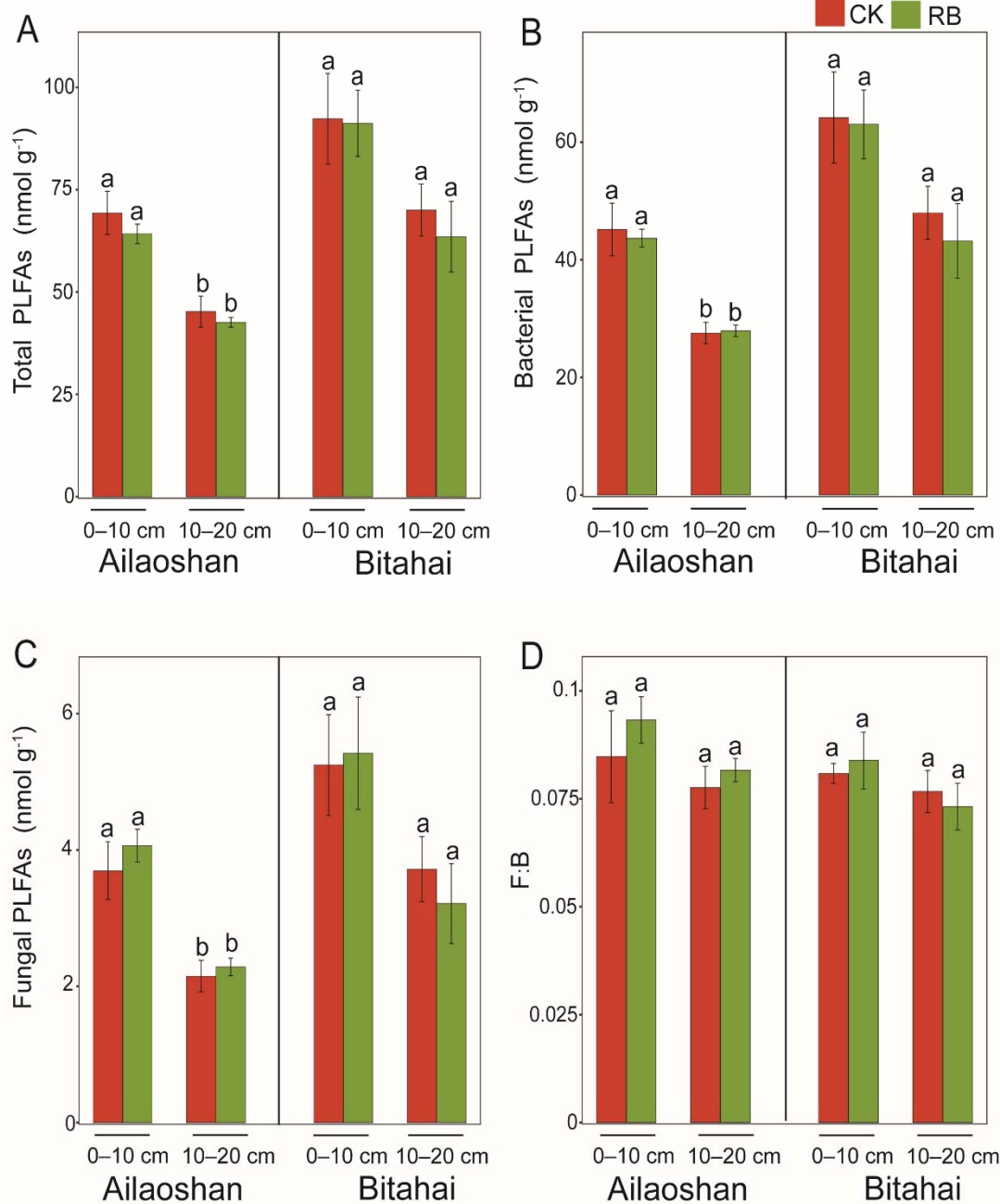

**Figure 3.** Effects of bryophyte removal (CK: bryophytes not removed; BR: bryophytes removed) and soil depth (0–10 and 10–20 cm) on soil microbial PLFAs in Ailaoshan and Bitahaiu. (**A**): total PLFAs, (**B**): bacterial PLFAs, (**C**): bacterial PLFAs, (**D**): the ratio of fungal and bacterial PLFAs. Values are the means $\pm$ SE of six plots. Different lowercase letters mean significant differences at $P < 0.05$ between treatments.

### 3.2. Effect of Bryophyte Removal on Soil Microbial Communities

In the evergreen forest, bryophyte removal significantly decreased the Chao1 index in the 0–10 cm layer (Figure 4A). Bryophyte removal had no significant effect on the Shannon index (Figure 4B). Two-way ANOVA showed soil depth significantly decreased the Chao1 index (Table 1). In the coniferous forest, bryophyte removal had no significant effect on the Chao1 index (Figure 4A) or the Shannon index (Figure 4B). A two-way ANOVA showed bryophyte removal decreased Chao1 index (Table 1). Three-way ANOVAs showed Chao1 index was significantly influenced by the interaction of bryophyte removal, soil

depth, and forest type, and interaction of soil depth and forest type (Table A1). In the evergreen forest, SOC ($P$ = 0.009) and C:N ($P$ = 0.025) were significantly and negatively correlated with microbial community composition, while SD ($P$ = 0.024) was significantly and positively correlated with microbial community composition (Figure 4C; Table A2). In the coniferous forest, SOC ($P$ = 0.001) and C:N ($P$ = 0.002) were negatively correlated with microbial community composition. TN ($P$ = 0.001), SBD ($P$ = 0.003), and SD ($P$ = 0.002) were positively correlated with microbial community composition (Figure 4D; Table A2).

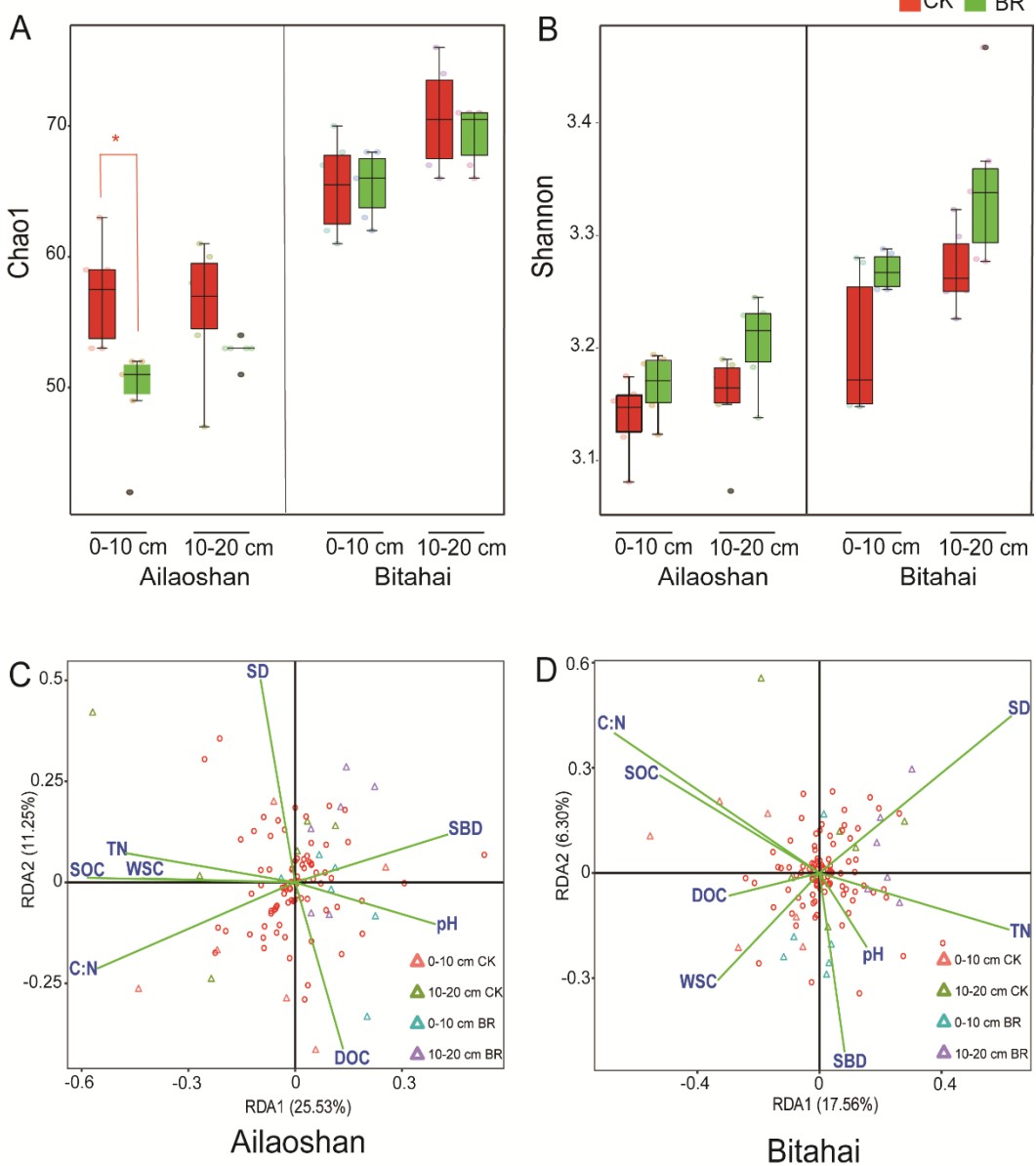

**Figure 4.** Effects of bryophyte removal (CK: no bryophyte removal; BR: bryophytes removed) and soil depth (0–10 and 10–20 cm) on soil microbial alpha diversity in Ailaoshan (**A**) and Bitahai (**B**). The red star in A means significant difference between treatments. Values are the means ± SE of six plots. Redundancy analysis (RDA) was used to study the relationship between soil physicochemical properties and microbial community composition in Ailaoshan (**C**) and Bitahai (**D**), the red circles represent the PLFA of microorganisms, the green lines represent physical and chemical factors, and the sample groups are represented by colored triangles.

### 3.3. Soil Organic Carbon Stock and Regulation Factors

In the evergreen forest, the total explanation rate of soil physicochemical properties and microbial changes in SOCs was 76.78%. TN, SOC, pH, and SD contributed more to SOCs changes, followed by soil microbial PLFAs, SBD, and bryophyte removal (Figure 5A). In the coniferous forest, the total explanation rate of soil physicochemical properties and microbial changes in SOCs was 43.12%, and only SOC and SD had significant effects on SOCs (Figure 5B).

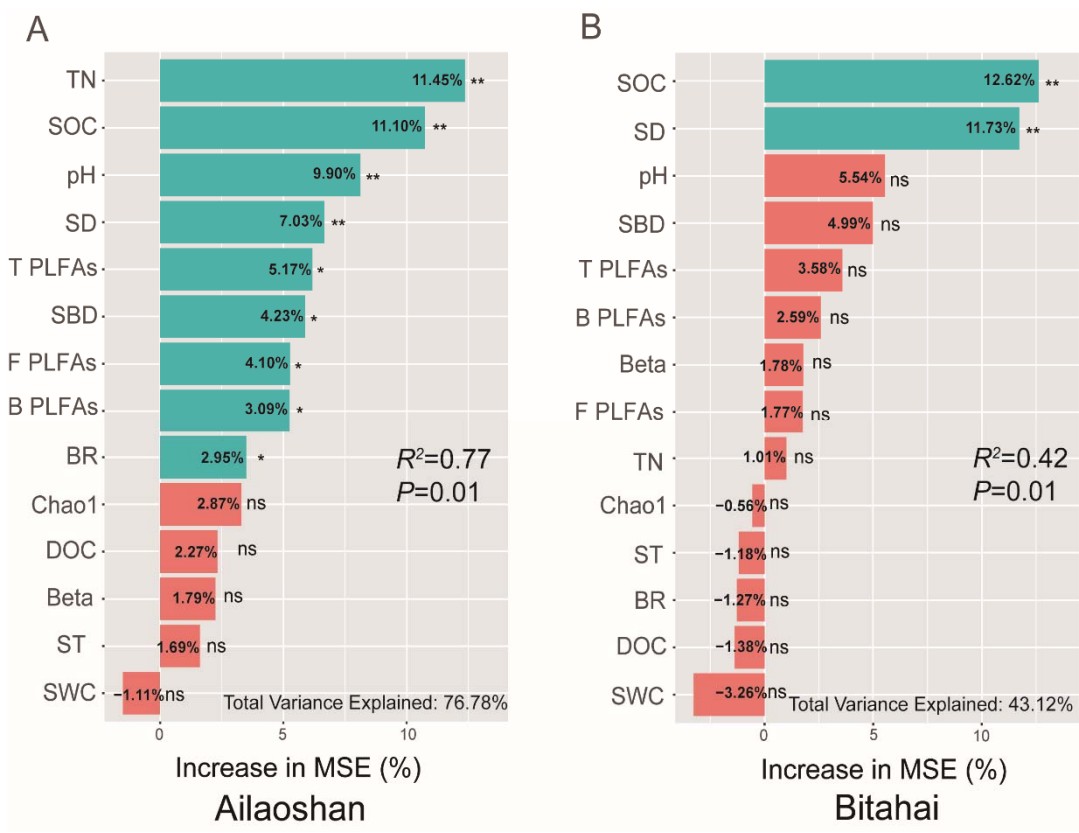

**Figure 5.** Effects of various factors on soil organic carbon stocks in Ailaoshan (**A**) and Bitahai (**B**) with random forest model. Factors in blue (*: $P < 0.05$, **: $P < 0.01$) represent significant contribution to soil organic carbon stocks, while factors in red (ns) represent insignificant contribution. BR: bryophyte removal, SOC: soil organic carbon, DOC: soil dissolved organic carbon, TN: soil total nitrogen, SWC: soil water content, SBD: soil bulk density, ST: surface temperature, SD: soil depth, T PLFAs: total microbial PLFAs, F PLFAs: fungal PLFAs, B PLFAs: bacterial PLFAs, Chao1: microbial Chao1 index, Beta: microbial beta diversity, Increase in MSE (%): percentage of increase of mean square error.

## 4. Discussion

### 4.1. Effect of Bryophyte Removal on Soil Organic Carbon Stock in Evergreen Broad-Leaved Forest

In the evergreen forest, bryophyte removal increased soil organic carbon stock, which was inconsistent with our first hypothesis. Previous studies had found that bryophyte mulch promotes the accumulation of soil organic carbon and nitrogen, because bryophytes can fix large amounts of carbon and nitrogen, especially the inputs from dead bryophytes [16,52,53]. The presence of bryophytes can also inhibit the decomposition of plant litter on the soil surface by producing secondary secretions, which facilitate soil carbon accumulation after bryophyte dies [54]. Our results were inconsistent with previous studies. The reason would be that removal of bryophytes decreased the plant carbon and nutrient input, consequently increasing soil bulk density. Soil organic carbon stock is closely related to soil bulk density [49], removal of ground bryophytes significantly increased soil bulk density could result in the increased soil organic carbon stock in our study.

Bryophytes can provide a carbon source for the growth of heterotrophic microorganisms [55]. Bryophyte removal could reduce microbial biomass and change microbial community structure. However, the microbial biomass, indicated by PLFAs, had no response to bryophyte removal. The insignificant effects on the microbial biomass resulted in unchanged soil organic carbon content in the evergreen forest. In addition, bryophyte can decrease soil pH value [56,57] and maintain the surface soil temperature [25]. Our results also showed that bryophyte removal increased soil temperature, which may affect soil organic carbon and nitrogen cycling [58,59]. When *S. junghuhnianum* was removed from the evergreen broad-leaved forest, the surface temperature increased by 1.1 °C. Consequently, this could affect soil organic carbon by altering greenhouse gas emissions and microbial activity in the soil [41]. The Chao1 index in the surface soil decreased, indicating decreased microbial diversity after bryophyte removal.

*4.2. Effect of Bryophyte Removal on Soil Organic Carbon Stock in Cold Temperate Coniferous Forest*

In the coniferous forest, bryophyte removal decreased soil organic carbon and soil organic carbon stock, which was consistent with our hypothesis. Bryophytes influence organic matter decomposition by regulating soil surface temperature and humidity [40,60], fixing atmospheric carbon through photosynthesis, and depositing carbon in soil [52]. Plant diversity in the coniferous forest was lower than in the evergreen forest in our study sites [44,46]. Thus, the effects of bryophyte removal in the evergreen forest would be stronger and decrease soil organic carbon stocks in the coniferous forest [17].

It is worth noting that bryophyte removal increased the surface soil nitrogen content in temperate coniferous forest. Previous studies suggested that bryophytes promoted the nitrogen cycle and can use the soil's available nitrogen [40], thus bryophyte removal would be beneficial for soil nitrogen availability. Other studies found that declines in bryophyte communities reduced soil organic carbon accumulation by decreasing nitrogen assimilation [28,61], which supported our results. Understory plant removal directly decreases ground plant cover, which usually leads to changes in soil physicochemical properties, thereby affecting soil microbial abundance and community structure [11,12]. Our results also show that the removal of bryophyte enhanced the Chao1 index based on two-way ANOVA. Increased microbial diversity could promote litter decomposition and negatively affect soil organic carbon content [30].

Significant differences in soil organic carbon, pH, soil water content, microbial diversity, and microbial PLFAs were found between the two forests. The results supported our second hypothesis that the subtropical evergreen broad-leaved forests are less responsive to bryophyte removal than cold temperate coniferous forests. In particular, the interaction of forest type and bryophyte removal had a significant effect on soil organic carbon stock, which further proved the important role of bryophyte in different forest types [62]. In addition, different soil depths have different effects on soil organic carbon, soil nitrogen, pH, SWC, and soil organic carbon stock, so the removal of bryophytes has a different response at different soil depths. Due to the decreases in microbial diversity and abundance with increasing soil depth, the effects of bryophyte removal may have been delayed in deeper soils [63].

**5. Conclusions**

Our study investigated the functions of bryophytes by removing bryophytes in both an evergreen broad-leaved forest and a cold temperate coniferous forest. We first found that bryophyte removal increased soil bulk density, soil temperature, and soil organic carbon stocks in the evergreen forest. However, bryophyte removal significantly decreased soil organic carbon and microbial diversity, but increased soil nitrogen in the coniferous forest. The responses of soil organic carbon stocks to bryophyte removal in the two forests were different. Second, the main drivers of soil characteristics were contrasted in each forest type after bryophyte removal, which in turn would affect soil microbial activity and carbon

and nitrogen cycles. Given the intensity of anthropogenic activities and global change, long-term and larger-scale studies are needed to explore the processes and consequences after the losses of bryophyte in forest ecosystems.

**Author Contributions:** J.W. conceived the study, all authors collected the data, performed statistical analysis, and wrote the manuscript. All authors have read and agreed to the published version of the manuscript.

**Funding:** This research was funded by National Natural Science Foundation of China (No. 31971497), the Project for Talent and Platform of Science and Technology in Yunnan Province Science and Technology Department (202205AM070005), the Xingdian Scholar Fund of Yunnan, Double Top University Plan Fund of Yunnan University, Yunnan Science and Technology Talent and Platform Program (202105AG070002) and the Postgraduate Research Innovation Fund of Yunnan University (2021Y045).

**Data Availability Statement:** Not applicable.

**Acknowledgments:** The authors thank Zhiyun Lu and other researchers and workers at Ailaoshan Station in Subtropical Forest Ecosystem Studies, Bitahai Nature Reserve and Baima Snow Mountain Ecological Field Observation and Research Station for field work assistance. We would like to thank Annalise Elliot at the University of Kansas for her assistance with English language and grammatical editing of the manuscript.

**Conflicts of Interest:** The authors declare no conflict of interest.

## Appendix A

**Table A1.** Bryophyte removal, soil depth, and forest type were used for three-way ANOVA.

| | | Bryophyte Removal (BR) | Forest Type (FT) | Soil Depth (SD) | BR × FT | BR × SD | FT × SD | BR × FT × SD |
|---|---|---|---|---|---|---|---|---|
| SOC | *F* | 4.30 | 9.94 | 129.15 | 1.73 | 1.15 | 0.03 | 0.06 |
| | *P* | 0.045 | 0.003 | 0.001 | 0.20 | 0.29 | 0.86 | 0.80 |
| SBD | *F* | 5.62 | 7.003 | 10.40 | 0.29 | 0.03 | 0.33 | 0.51 |
| | *P* | 0.02 | 0.012 | 0.003 | 0.60 | 0.86 | 0.57 | 0.48 |
| SOCs | *F* | 0.09 | 29.07 | 80.95 | 5.68 | 0.03 | 1.03 | 0.91 |
| | *P* | 0.77 | 0.001 | 0.001 | 0.02 | 0.87 | 0.32 | 0.35 |
| DOC | *F* | 2.18 | 0.24 | 3.87 | 0.23 | 0.14 | 0.01 | 1.21 |
| | *P* | 0.15 | 0.63 | 0.06 | 0.64 | 0.71 | 0.93 | 0.28 |
| TN | *F* | 0.73 | 2.27 | 14.27 | 2.34 | 3.99 | 53.40 | 6.43 |
| | *P* | 0.40 | 0.14 | 0.001 | 0.13 | 0.15 | <0.001 | 0.02 |
| pH | *F* | 0.41 | 84.31 | 24.94 | 0.72 | 2.04 | 0.02 | 0.67 |
| | *P* | 0.53 | 0.001 | <0.001 | 0.40 | 0.16 | 0.89 | 0.42 |
| SWC | *F* | 0.07 | 12.89 | 2.94 | 1.30 | 0.03 | 1.71 | 0.24 |
| | *P* | 0.80 | <0.001 | 0.09 | 0.26 | 0.86 | 0.20 | 0.63 |
| T PLFAs | *F* | 0.67 | 26.01 | 25.95 | <0.001 | 0.03 | 0.05 | 0.18 |
| | *P* | 0.42 | <0.001 | <0.001 | 0.99 | 0.87 | 0.82 | 0.68 |
| B PLFAs | *F* | 0.27 | 30.22 | 26.52 | 0.13 | 0.02 | 0.04 | 0.17 |
| | *P* | 0.61 | <0.001 | <0.001 | 0.72 | 0.90 | 0.84 | 0.68 |
| F PLFAs | *F* | 0.01 | 13.89 | 23.68 | 0.33 | 0.39 | 0.08 | 0.09 |
| | *P* | 0.91 | <0.001 | <0.001 | 0.57 | 0.53 | 0.78 | 0.76 |
| Chao1 | *F* | 7.95 | 189.52 | 8.91 | 3.03 | 0.56 | 5.78 | 2.23 |
| | *P* | 0.007 | <0.001 | 0.005 | 0.09 | 0.46 | 0.02 | 0.14 |
| Beta | *F* | 0.56 | 0.61 | 0.02 | 3.24 | 2.07 | 0.06 | 0.61 |
| | *P* | 0.46 | 0.44 | 0.87 | 0.08 | 0.16 | 0.81 | 0.44 |

Note: *F* and *P* values for the effects of bryophyte removal (BR), forest type (FT), and soil depth (SD) and their interactions on soil physicochemical properties and soil microorganisms. SOC: soil organic carbon, DOC: soil dissolved organic carbon, TN: soil total nitrogen, SOCs: soil organic carbon stocks, SWC: soil water content, SBD: soil bulk density, T PLFAs: total microbial PLFAs, F PLFAs: fungal PLFAs, B PLFAs: bacterial PLFAs, Chao1: microbial Chao1 index, Beta: microbial beta diversity.

**Table A2.** Redundancy analysis statistics.

| Indication | ALS | | | | BTH | | | |
|---|---|---|---|---|---|---|---|---|
| | RDA1 | RDA2 | $r^2$ | $P$ | RDA1 | RDA2 | $r^2$ | $P$ |
| SOC | −0.95 | −0.31 | 0.32 | 0.01 | −0.65 | −0.76 | 0.49 | 0.001 |
| DOC | −0.09 | 0.10 | 0.09 | 0.40 | −0.76 | −0.65 | 0.10 | 0.34 |
| TN | −0.94 | −0.35 | 0.23 | 0.08 | 0.82 | 0.57 | 0.45 | 0.002 |
| C:N | −0.99 | −0.14 | 0.30 | 0.03 | −0.69 | −0.73 | 0.64 | 0.001 |
| pH | 0.89 | 0.45 | 0.19 | 0.11 | 0.27 | 0.96 | 0.23 | 0.07 |
| SWC | −0.94 | −0.34 | 0.15 | 0.18 | −0.99 | 0.07 | 0.09 | 0.39 |
| SBD | 0.98 | 0.19 | 0.18 | 0.12 | 0.14 | 0.99 | 0.41 | 0.003 |
| SD | 0.05 | −0.99 | 0.23 | 0.02 | 0.66 | −0.75 | 0.56 | 0.002 |

Note: RDA1 indicates the cosine value of the angle between soil factor and ordination axis, and RDA2 indicates the correlation between soil factor and ordination axis. $r^2$ is the determination coefficient of soil factors in microbial distribution. The smaller $r^2$, the smaller the influence of soil factors on species distribution. $P$ is the significance test of correlation, $P < 0.05$ represents a significant. SOC: soil organic carbon, DOC: soil dissolved organic carbon, TN: soil total nitrogen, SWC: soil water content, SBD: soil bulk density, C:N: soil C/N ratio, SD: soil depth.

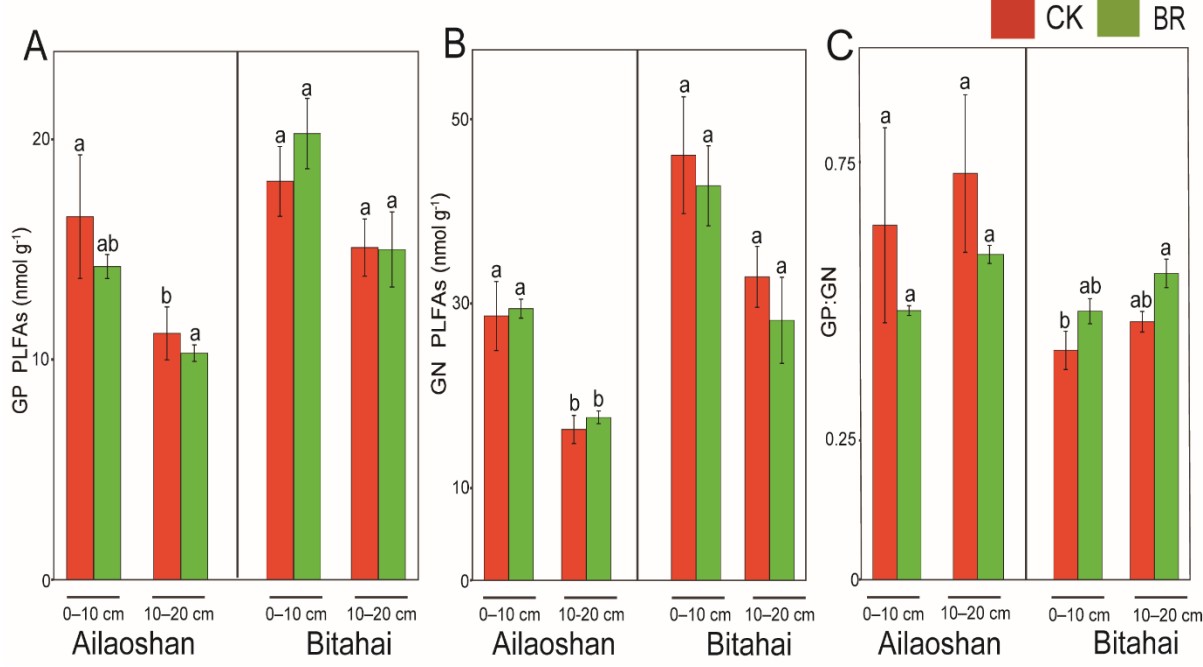

**Figure A1.** Effects of bryophyte removal (CK: bryophytes not removed, BR: bryophyte removed) and soil depth (0–10 and 10–20 cm) on soil microbial PLFAs in Ailaoshan and Bitahai. (**A**): GP PLFAs: Gram-positive bacterial PLFAs, (**B**): GN PLFAs: Gram-negative bacterial PLFAs, (**C**): GP:GN: ratio of Gram-positive bacterial PLFAs to Gram-negative bacterial PLFAs. Values are the means ± SE of six plots. Different lowercase letters mean significant differences at $P < 0.05$ between treatments.

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
