# Peer review of "Response of Soil Organic Carbon Stock to Bryophyte Removal Is Regulated by Forest Types in Southwest China"

_forests, doi:10.3390/f13122125_

Round 1

Reviewer 1 Report

Review of the ms ‘Response of soil organic carbon stock to bryophyte removal……..’ by Chen et al. Ms ID Forests – 2010323

General comments

The authors have submitted a potentially interesting ms on the effects on the forest soil when bryophytes are removed. They used two forests; a subtropical one with evergreen broadleaf trees and a Sphagnum species as dominant bryophyte; the other forest was a temperate coniferous spruce and fir forest with a moss of the genus Actinothuidium.  Each investigation area was divided into six blocks each (6 control and 6 with the bryophytes removed). The experiment started in May 2017 and a sampling took place one year later. In the two forests the authors measured soil C and N as well as microbial biomass indicators. The authors conclude that the removal of these bryophytes has a clear influence on the sequestration of C and N in the forest floor. The study is interesting and could possibly be published after a rewriting. There is a number of problems with the present ms - please see below, and the English needs to be edited. I recommend a that the ms is rejected and that a resubmission is encouraged.

Specific comments

My major concern is the Methods’ section

The general design is very unclear. The authors removed the C- and N-containing bryophyte mass and measured the effects on the soil (apparently the mineral soil). Did the authors do this just once or did they repeat the removal? Did they remove the live parts of the plants or live parts plus attached dead – with e.g. Sphagnum this makes a clear difference. The authors investigated the soil after one year – as far as I can see they may have found a set of differences but I cannot find any explanation. In line 167 the authors claim that removal of bryophytes increased soil organic carbon stock (SOCs), which intuitively is contradictory as a source of C was removed. Still, I cannot find any explanation. Measurements of microbial biomass is interesting per se but does not explain any change in C and N.  Details below.  

In section 2.3 Soil measurements - the authors have not given any number for repetitions. They sampled for bulk density and I assume that they sampled the mineral soil; what depths? How many replicates. Do the authors refer to the 3 soil cores mentioned in line 121? These were sampled with one sample per plot. Is ‘plot’ the same as a block? This should be written more clearly.

Line 128 Soil water content was sampled – how many replicates? With what intensity - how often? Soil temperature - how many probes and for how long? How many replicates of soil pH? How many replicates of samples for soil C and N?  The analysis of C and N could be given in more detail? Did the authors need to separate organic and inorganic C? What digestion method was used?

Some more detail about the analysis for phospholipid fatty acids analysis is needed. The authors could also at least briefly mention what the Chao1 and Shannon indices are and what they tell the reader.

One piece of information is the duration of the experiment. As the authors give the effects of bryophyte removal on e.g. soil carbon and nitrogen – was this an experiment with just one removal of bryophyte and measurements made once a year after?

In the statistical analysis – how did the authors compare the effects at the two sites?

The Results and Discussion sections are in parts difficult to follow; some comments are given below.

Table 1. There are six columns giving F- and P- values. Some of the p values have a less-than–sign (<) others not and in some cases there should have been, I think, the sign =. Is that correct?

Figures 2 and 3. The significances given in the upper right corner of each subfigure – what do they say?

Figure 4 ‘Different lower-case letters mean……’  I cannot find any lower-case letters at all. Please add that or change the figure.  Subfigures C and D – what do they show? The legend needs to be developed.

Figure 5. Please define MSE.  I searched but could not find what the abbreviation stands for.

Lines 235. The sentence is a bit enigmatic. Please develop and explain so the reader can understand how the statement fits into the context. In fact, the whole first para (lines 233 thru 240) needs to be developed. The authors need to explain how litter decomposition rate has an effect.

Line 258. How does plant diversity come into this study? The authors did not study that and give no reference to their statement.

Line 262.  ‘nitrogen conversion’ ? Conversion to what?

The lines 261 to 265 appear enigmatic – the only thing they tell me is that there are different results reported but the authors do not suggest any reason to that.

Some details

Lines 89 and 80. Please mention what the names stand for.

Line 60. That reference appears to give a study in the Antarctica. The quantitative photosynthesis by a sphagnum there may not be compared to that is a subtropical forest- or?

Line 115. Reomval should be removal.

Line 132 SOC has already been defined.

Line 143 what does ‘relative nmol per g dry soil mean’? Relative nmol?

Line 166 Fig 2G should be Fig 2F

Line 164 Delete ‘other’Review of the ms ‘Response of soil organic carbon stock to bryophyte removal……..’ by Chen et al. Ms ID Forests – 2010323

General comments

The authors have submitted a potentially interesting ms on the effects on the forest soil when bryophytes are removed. They used two forests; a subtropical one with evergreen broadleaf trees and a Sphagnum species as dominant bryophyte; the other forest was a temperate coniferous spruce and fir forest with a moss of the genus Actinothuidium.  Each investigation area was divided into six blocks each (6 control and 6 with the bryophytes removed). The experiment started in May 2017 and a sampling took place one year later. In the two forests the authors measured soil C and N as well as microbial biomass indicators. The authors conclude that the removal of these bryophytes has a clear influence on the sequestration of C and N in the forest floor. The study is interesting and could possibly be published after a rewriting. There is a number of problems with the present ms - please see below, and the English needs to be edited. I recommend a that the ms is rejected and that a resubmission is encouraged.

Specific comments

My major concern is the Methods’ section

The general design is very unclear. The authors removed the C- and N-containing bryophyte mass and measured the effects on the soil (apparently the mineral soil). Did the authors do this just once or did they repeat the removal? Did they remove the live parts of the plants or live parts plus attached dead – with e.g. Sphagnum this makes a clear difference. The authors investigated the soil after one year – as far as I can see they may have found a set of differences but I cannot find any explanation. In line 167 the authors claim that removal of bryophytes increased soil organic carbon stock (SOCs), which intuitively is contradictory as a source of C was removed. Still, I cannot find any explanation. Measurements of microbial biomass is interesting per se but does not explain any change in C and N.  Details below.  

In section 2.3 Soil measurements - the authors have not given any number for repetitions. They sampled for bulk density and I assume that they sampled the mineral soil; what depths? How many replicates. Do the authors refer to the 3 soil cores mentioned in line 121? These were sampled with one sample per plot. Is ‘plot’ the same as a block? This should be written more clearly.

Line 128 Soil water content was sampled – how many replicates? With what intensity - how often? Soil temperature - how many probes and for how long? How many replicates of soil pH? How many replicates of samples for soil C and N?  The analysis of C and N could be given in more detail? Did the authors need to separate organic and inorganic C? What digestion method was used?

Some more detail about the analysis for phospholipid fatty acids analysis is needed. The authors could also at least briefly mention what the Chao1 and Shannon indices are and what they tell the reader.

One piece of information is the duration of the experiment. As the authors give the effects of bryophyte removal on e.g. soil carbon and nitrogen – was this an experiment with just one removal of bryophyte and measurements made once a year after?

In the statistical analysis – how did the authors compare the effects at the two sites?

The Results and Discussion sections are in parts difficult to follow; some comments are given below.

Table 1. There are six columns giving F- and P- values. Some of the p values have a less-than–sign (<) others not and in some cases there should have been, I think, the sign =. Is that correct?

Figures 2 and 3. The significances given in the upper right corner of each subfigure – what do they say?

Figure 4 ‘Different lower-case letters mean……’  I cannot find any lower-case letters at all. Please add that or change the figure.  Subfigures C and D – what do they show? The legend needs to be developed.

Figure 5. Please define MSE.  I searched but could not find what the abbreviation stands for.

Lines 235. The sentence is a bit enigmatic. Please develop and explain so the reader can understand how the statement fits into the context. In fact, the whole first para (lines 233 thru 240) needs to be developed. The authors need to explain how litter decomposition rate has an effect.

Line 258. How does plant diversity come into this study? The authors did not study that and give no reference to their statement.

Line 262.  ‘nitrogen conversion’ ? Conversion to what?

The lines 261 to 265 appear enigmatic – the only thing they tell me is that there are different results reported but the authors do not suggest any reason to that.

Some details

Lines 89 and 80. Please mention what the names stand for.

Line 60. That reference appears to give a study in the Antarctica. The quantitative photosynthesis by a sphagnum there may not be compared to that is a subtropical forest- or?

Line 115. Reomval should be removal.

Line 132 SOC has already been defined.

Line 143 what does ‘relative nmol per g dry soil mean’? Relative nmol?

Line 166 Fig 2G should be Fig 2F

Line 164 Delete ‘other’Review of the ms ‘Response of soil organic carbon stock to bryophyte removal……..’ by Chen et al. Ms ID Forests – 2010323

General comments

The authors have submitted a potentially interesting ms on the effects on the forest soil when bryophytes are removed. They used two forests; a subtropical one with evergreen broadleaf trees and a Sphagnum species as dominant bryophyte; the other forest was a temperate coniferous spruce and fir forest with a moss of the genus Actinothuidium.  Each investigation area was divided into six blocks each (6 control and 6 with the bryophytes removed). The experiment started in May 2017 and a sampling took place one year later. In the two forests the authors measured soil C and N as well as microbial biomass indicators. The authors conclude that the removal of these bryophytes has a clear influence on the sequestration of C and N in the forest floor. The study is interesting and could possibly be published after a rewriting. There is a number of problems with the present ms - please see below, and the English needs to be edited. I recommend a that the ms is rejected and that a resubmission is encouraged.

Specific comments

My major concern is the Methods’ section

The general design is very unclear. The authors removed the C- and N-containing bryophyte mass and measured the effects on the soil (apparently the mineral soil). Did the authors do this just once or did they repeat the removal? Did they remove the live parts of the plants or live parts plus attached dead – with e.g. Sphagnum this makes a clear difference. The authors investigated the soil after one year – as far as I can see they may have found a set of differences but I cannot find any explanation. In line 167 the authors claim that removal of bryophytes increased soil organic carbon stock (SOCs), which intuitively is contradictory as a source of C was removed. Still, I cannot find any explanation. Measurements of microbial biomass is interesting per se but does not explain any change in C and N.  Details below.  

In section 2.3 Soil measurements - the authors have not given any number for repetitions. They sampled for bulk density and I assume that they sampled the mineral soil; what depths? How many replicates. Do the authors refer to the 3 soil cores mentioned in line 121? These were sampled with one sample per plot. Is ‘plot’ the same as a block? This should be written more clearly.

Line 128 Soil water content was sampled – how many replicates? With what intensity - how often? Soil temperature - how many probes and for how long? How many replicates of soil pH? How many replicates of samples for soil C and N?  The analysis of C and N could be given in more detail? Did the authors need to separate organic and inorganic C? What digestion method was used?

Some more detail about the analysis for phospholipid fatty acids analysis is needed. The authors could also at least briefly mention what the Chao1 and Shannon indices are and what they tell the reader.

One piece of information is the duration of the experiment. As the authors give the effects of bryophyte removal on e.g. soil carbon and nitrogen – was this an experiment with just one removal of bryophyte and measurements made once a year after?

In the statistical analysis – how did the authors compare the effects at the two sites?

The Results and Discussion sections are in parts difficult to follow; some comments are given below.

Table 1. There are six columns giving F- and P- values. Some of the p values have a less-than–sign (<) others not and in some cases there should have been, I think, the sign =. Is that correct?

Figures 2 and 3. The significances given in the upper right corner of each subfigure – what do they say?

Figure 4 ‘Different lower-case letters mean……’  I cannot find any lower-case letters at all. Please add that or change the figure.  Subfigures C and D – what do they show? The legend needs to be developed.

Figure 5. Please define MSE.  I searched but could not find what the abbreviation stands for.

Lines 235. The sentence is a bit enigmatic. Please develop and explain so the reader can understand how the statement fits into the context. In fact, the whole first para (lines 233 thru 240) needs to be developed. The authors need to explain how litter decomposition rate has an effect.

Line 258. How does plant diversity come into this study? The authors did not study that and give no reference to their statement.

Line 262.  ‘nitrogen conversion’ ? Conversion to what?

The lines 261 to 265 appear enigmatic – the only thing they tell me is that there are different results reported but the authors do not suggest any reason to that.

Some details

Lines 89 and 80. Please mention what the names stand for.

Line 60. That reference appears to give a study in the Antarctica. The quantitative photosynthesis by a sphagnum there may not be compared to that is a subtropical forest- or?

Line 115. Reomval should be removal.

Line 132 SOC has already been defined.

Line 143 what does ‘relative nmol per g dry soil mean’? Relative nmol?

Line 166 Fig 2G should be Fig 2F

Line 164 Delete ‘other’Review of the ms ‘Response of soil organic carbon stock to bryophyte removal……..’ by Chen et al. Ms ID Forests – 2010323

General comments

The authors have submitted a potentially interesting ms on the effects on the forest soil when bryophytes are removed. They used two forests; a subtropical one with evergreen broadleaf trees and a Sphagnum species as dominant bryophyte; the other forest was a temperate coniferous spruce and fir forest with a moss of the genus Actinothuidium.  Each investigation area was divided into six blocks each (6 control and 6 with the bryophytes removed). The experiment started in May 2017 and a sampling took place one year later. In the two forests the authors measured soil C and N as well as microbial biomass indicators. The authors conclude that the removal of these bryophytes has a clear influence on the sequestration of C and N in the forest floor. The study is interesting and could possibly be published after a rewriting. There is a number of problems with the present ms - please see below, and the English needs to be edited. I recommend a that the ms is rejected and that a resubmission is encouraged.

Specific comments

My major concern is the Methods’ section

The general design is very unclear. The authors removed the C- and N-containing bryophyte mass and measured the effects on the soil (apparently the mineral soil). Did the authors do this just once or did they repeat the removal? Did they remove the live parts of the plants or live parts plus attached dead – with e.g. Sphagnum this makes a clear difference. The authors investigated the soil after one year – as far as I can see they may have found a set of differences but I cannot find any explanation. In line 167 the authors claim that removal of bryophytes increased soil organic carbon stock (SOCs), which intuitively is contradictory as a source of C was removed. Still, I cannot find any explanation. Measurements of microbial biomass is interesting per se but does not explain any change in C and N.  Details below.  

In section 2.3 Soil measurements - the authors have not given any number for repetitions. They sampled for bulk density and I assume that they sampled the mineral soil; what depths? How many replicates. Do the authors refer to the 3 soil cores mentioned in line 121? These were sampled with one sample per plot. Is ‘plot’ the same as a block? This should be written more clearly.

Line 128 Soil water content was sampled – how many replicates? With what intensity - how often? Soil temperature - how many probes and for how long? How many replicates of soil pH? How many replicates of samples for soil C and N?  The analysis of C and N could be given in more detail? Did the authors need to separate organic and inorganic C? What digestion method was used?

Some more detail about the analysis for phospholipid fatty acids analysis is needed. The authors could also at least briefly mention what the Chao1 and Shannon indices are and what they tell the reader.

One piece of information is the duration of the experiment. As the authors give the effects of bryophyte removal on e.g. soil carbon and nitrogen – was this an experiment with just one removal of bryophyte and measurements made once a year after?

In the statistical analysis – how did the authors compare the effects at the two sites?

The Results and Discussion sections are in parts difficult to follow; some comments are given below.

Table 1. There are six columns giving F- and P- values. Some of the p values have a less-than–sign (<) others not and in some cases there should have been, I think, the sign =. Is that correct?

Figures 2 and 3. The significances given in the upper right corner of each subfigure – what do they say?

Figure 4 ‘Different lower-case letters mean……’  I cannot find any lower-case letters at all. Please add that or change the figure.  Subfigures C and D – what do they show? The legend needs to be developed.

Figure 5. Please define MSE.  I searched but could not find what the abbreviation stands for.

Lines 235. The sentence is a bit enigmatic. Please develop and explain so the reader can understand how the statement fits into the context. In fact, the whole first para (lines 233 thru 240) needs to be developed. The authors need to explain how litter decomposition rate has an effect.

Line 258. How does plant diversity come into this study? The authors did not study that and give no reference to their statement.

Line 262.  ‘nitrogen conversion’ ? Conversion to what?

The lines 261 to 265 appear enigmatic – the only thing they tell me is that there are different results reported but the authors do not suggest any reason to that.

Some details

Lines 89 and 80. Please mention what the names stand for.

Line 60. That reference appears to give a study in the Antarctica. The quantitative photosynthesis by a sphagnum there may not be compared to that is a subtropical forest- or?

Line 115. Reomval should be removal.

Line 132 SOC has already been defined.

Line 143 what does ‘relative nmol per g dry soil mean’? Relative nmol?

Line 166 Fig 2G should be Fig 2F

Line 164 Delete ‘other’Review of the ms ‘Response of soil organic carbon stock to bryophyte removal……..’ by Chen et al. Ms ID Forests – 2010323

General comments

The authors have submitted a potentially interesting ms on the effects on the forest soil when bryophytes are removed. They used two forests; a subtropical one with evergreen broadleaf trees and a Sphagnum species as dominant bryophyte; the other forest was a temperate coniferous spruce and fir forest with a moss of the genus Actinothuidium.  Each investigation area was divided into six blocks each (6 control and 6 with the bryophytes removed). The experiment started in May 2017 and a sampling took place one year later. In the two forests the authors measured soil C and N as well as microbial biomass indicators. The authors conclude that the removal of these bryophytes has a clear influence on the sequestration of C and N in the forest floor. The study is interesting and could possibly be published after a rewriting. There is a number of problems with the present ms - please see below, and the English needs to be edited. I recommend a that the ms is rejected and that a resubmission is encouraged.

Specific comments

My major concern is the Methods’ section

The general design is very unclear. The authors removed the C- and N-containing bryophyte mass and measured the effects on the soil (apparently the mineral soil). Did the authors do this just once or did they repeat the removal? Did they remove the live parts of the plants or live parts plus attached dead – with e.g. Sphagnum this makes a clear difference. The authors investigated the soil after one year – as far as I can see they may have found a set of differences but I cannot find any explanation. In line 167 the authors claim that removal of bryophytes increased soil organic carbon stock (SOCs), which intuitively is contradictory as a source of C was removed. Still, I cannot find any explanation. Measurements of microbial biomass is interesting per se but does not explain any change in C and N.  Details below.  

In section 2.3 Soil measurements - the authors have not given any number for repetitions. They sampled for bulk density and I assume that they sampled the mineral soil; what depths? How many replicates. Do the authors refer to the 3 soil cores mentioned in line 121? These were sampled with one sample per plot. Is ‘plot’ the same as a block? This should be written more clearly.

Line 128 Soil water content was sampled – how many replicates? With what intensity - how often? Soil temperature - how many probes and for how long? How many replicates of soil pH? How many replicates of samples for soil C and N?  The analysis of C and N could be given in more detail? Did the authors need to separate organic and inorganic C? What digestion method was used?

Some more detail about the analysis for phospholipid fatty acids analysis is needed. The authors could also at least briefly mention what the Chao1 and Shannon indices are and what they tell the reader.

One piece of information is the duration of the experiment. As the authors give the effects of bryophyte removal on e.g. soil carbon and nitrogen – was this an experiment with just one removal of bryophyte and measurements made once a year after?

In the statistical analysis – how did the authors compare the effects at the two sites?

The Results and Discussion sections are in parts difficult to follow; some comments are given below.

Table 1. There are six columns giving F- and P- values. Some of the p values have a less-than–sign (<) others not and in some cases there should have been, I think, the sign =. Is that correct?

Figures 2 and 3. The significances given in the upper right corner of each subfigure – what do they say?

Figure 4 ‘Different lower-case letters mean……’  I cannot find any lower-case letters at all. Please add that or change the figure.  Subfigures C and D – what do they show? The legend needs to be developed.

Figure 5. Please define MSE.  I searched but could not find what the abbreviation stands for.

Lines 235. The sentence is a bit enigmatic. Please develop and explain so the reader can understand how the statement fits into the context. In fact, the whole first para (lines 233 thru 240) needs to be developed. The authors need to explain how litter decomposition rate has an effect.

Line 258. How does plant diversity come into this study? The authors did not study that and give no reference to their statement.

Line 262.  ‘nitrogen conversion’ ? Conversion to what?

The lines 261 to 265 appear enigmatic – the only thing they tell me is that there are different results reported but the authors do not suggest any reason to that.

Some details

Lines 89 and 80. Please mention what the names stand for.

Line 60. That reference appears to give a study in the Antarctica. The quantitative photosynthesis by a sphagnum there may not be compared to that is a subtropical forest- or?

Line 115. Reomval should be removal.

Line 132 SOC has already been defined.

Line 143 what does ‘relative nmol per g dry soil mean’? Relative nmol?

Line 166 Fig 2G should be Fig 2F

Line 164 Delete ‘other’Review of the ms ‘Response of soil organic carbon stock to bryophyte removal……..’ by Chen et al. Ms ID Forests – 2010323

General comments

The authors have submitted a potentially interesting ms on the effects on the forest soil when bryophytes are removed. They used two forests; a subtropical one with evergreen broadleaf trees and a Sphagnum species as dominant bryophyte; the other forest was a temperate coniferous spruce and fir forest with a moss of the genus Actinothuidium.  Each investigation area was divided into six blocks each (6 control and 6 with the bryophytes removed). The experiment started in May 2017 and a sampling took place one year later. In the two forests the authors measured soil C and N as well as microbial biomass indicators. The authors conclude that the removal of these bryophytes has a clear influence on the sequestration of C and N in the forest floor. The study is interesting and could possibly be published after a rewriting. There is a number of problems with the present ms - please see below, and the English needs to be edited. I recommend a that the ms is rejected and that a resubmission is encouraged.

Specific comments

My major concern is the Methods’ section

The general design is very unclear. The authors removed the C- and N-containing bryophyte mass and measured the effects on the soil (apparently the mineral soil). Did the authors do this just once or did they repeat the removal? Did they remove the live parts of the plants or live parts plus attached dead – with e.g. Sphagnum this makes a clear difference. The authors investigated the soil after one year – as far as I can see they may have found a set of differences but I cannot find any explanation. In line 167 the authors claim that removal of bryophytes increased soil organic carbon stock (SOCs), which intuitively is contradictory as a source of C was removed. Still, I cannot find any explanation. Measurements of microbial biomass is interesting per se but does not explain any change in C and N.  Details below.  

In section 2.3 Soil measurements - the authors have not given any number for repetitions. They sampled for bulk density and I assume that they sampled the mineral soil; what depths? How many replicates. Do the authors refer to the 3 soil cores mentioned in line 121? These were sampled with one sample per plot. Is ‘plot’ the same as a block? This should be written more clearly.

Line 128 Soil water content was sampled – how many replicates? With what intensity - how often? Soil temperature - how many probes and for how long? How many replicates of soil pH? How many replicates of samples for soil C and N?  The analysis of C and N could be given in more detail? Did the authors need to separate organic and inorganic C? What digestion method was used?

Some more detail about the analysis for phospholipid fatty acids analysis is needed. The authors could also at least briefly mention what the Chao1 and Shannon indices are and what they tell the reader.

One piece of information is the duration of the experiment. As the authors give the effects of bryophyte removal on e.g. soil carbon and nitrogen – was this an experiment with just one removal of bryophyte and measurements made once a year after?

In the statistical analysis – how did the authors compare the effects at the two sites?

The Results and Discussion sections are in parts difficult to follow; some comments are given below.

Table 1. There are six columns giving F- and P- values. Some of the p values have a less-than–sign (<) others not and in some cases there should have been, I think, the sign =. Is that correct?

Figures 2 and 3. The significances given in the upper right corner of each subfigure – what do they say?

Figure 4 ‘Different lower-case letters mean……’  I cannot find any lower-case letters at all. Please add that or change the figure.  Subfigures C and D – what do they show? The legend needs to be developed.

Figure 5. Please define MSE.  I searched but could not find what the abbreviation stands for.

Lines 235. The sentence is a bit enigmatic. Please develop and explain so the reader can understand how the statement fits into the context. In fact, the whole first para (lines 233 thru 240) needs to be developed. The authors need to explain how litter decomposition rate has an effect.

Line 258. How does plant diversity come into this study? The authors did not study that and give no reference to their statement.

Line 262.  ‘nitrogen conversion’ ? Conversion to what?

The lines 261 to 265 appear enigmatic – the only thing they tell me is that there are different results reported but the authors do not suggest any reason to that.

Some details

Lines 89 and 80. Please mention what the names stand for.

Line 60. That reference appears to give a study in the Antarctica. The quantitative photosynthesis by a sphagnum there may not be compared to that is a subtropical forest- or?

Line 115. Reomval should be removal.

Line 132 SOC has already been defined.

Line 143 what does ‘relative nmol per g dry soil mean’? Relative nmol?

Line 166 Fig 2G should be Fig 2F

Line 164 Delete ‘other’Review of the ms ‘Response of soil organic carbon stock to bryophyte removal……..’ by Chen et al. Ms ID Forests – 2010323

General comments

The authors have submitted a potentially interesting ms on the effects on the forest soil when bryophytes are removed. They used two forests; a subtropical one with evergreen broadleaf trees and a Sphagnum species as dominant bryophyte; the other forest was a temperate coniferous spruce and fir forest with a moss of the genus Actinothuidium.  Each investigation area was divided into six blocks each (6 control and 6 with the bryophytes removed). The experiment started in May 2017 and a sampling took place one year later. In the two forests the authors measured soil C and N as well as microbial biomass indicators. The authors conclude that the removal of these bryophytes has a clear influence on the sequestration of C and N in the forest floor. The study is interesting and could possibly be published after a rewriting. There is a number of problems with the present ms - please see below, and the English needs to be edited. I recommend a that the ms is rejected and that a resubmission is encouraged.

Specific comments

My major concern is the Methods’ section

The general design is very unclear. The authors removed the C- and N-containing bryophyte mass and measured the effects on the soil (apparently the mineral soil). Did the authors do this just once or did they repeat the removal? Did they remove the live parts of the plants or live parts plus attached dead – with e.g. Sphagnum this makes a clear difference. The authors investigated the soil after one year – as far as I can see they may have found a set of differences but I cannot find any explanation. In line 167 the authors claim that removal of bryophytes increased soil organic carbon stock (SOCs), which intuitively is contradictory as a source of C was removed. Still, I cannot find any explanation. Measurements of microbial biomass is interesting per se but does not explain any change in C and N.  Details below.  

In section 2.3 Soil measurements - the authors have not given any number for repetitions. They sampled for bulk density and I assume that they sampled the mineral soil; what depths? How many replicates. Do the authors refer to the 3 soil cores mentioned in line 121? These were sampled with one sample per plot. Is ‘plot’ the same as a block? This should be written more clearly.

Line 128 Soil water content was sampled – how many replicates? With what intensity - how often? Soil temperature - how many probes and for how long? How many replicates of soil pH? How many replicates of samples for soil C and N?  The analysis of C and N could be given in more detail? Did the authors need to separate organic and inorganic C? What digestion method was used?

Some more detail about the analysis for phospholipid fatty acids analysis is needed. The authors could also at least briefly mention what the Chao1 and Shannon indices are and what they tell the reader.

One piece of information is the duration of the experiment. As the authors give the effects of bryophyte removal on e.g. soil carbon and nitrogen – was this an experiment with just one removal of bryophyte and measurements made once a year after?

In the statistical analysis – how did the authors compare the effects at the two sites?

The Results and Discussion sections are in parts difficult to follow; some comments are given below.

Table 1. There are six columns giving F- and P- values. Some of the p values have a less-than–sign (<) others not and in some cases there should have been, I think, the sign =. Is that correct?

Figures 2 and 3. The significances given in the upper right corner of each subfigure – what do they say?

Figure 4 ‘Different lower-case letters mean……’  I cannot find any lower-case letters at all. Please add that or change the figure.  Subfigures C and D – what do they show? The legend needs to be developed.

Figure 5. Please define MSE.  I searched but could not find what the abbreviation stands for.

Lines 235. The sentence is a bit enigmatic. Please develop and explain so the reader can understand how the statement fits into the context. In fact, the whole first para (lines 233 thru 240) needs to be developed. The authors need to explain how litter decomposition rate has an effect.

Line 258. How does plant diversity come into this study? The authors did not study that and give no reference to their statement.

Line 262.  ‘nitrogen conversion’ ? Conversion to what?

The lines 261 to 265 appear enigmatic – the only thing they tell me is that there are different results reported but the authors do not suggest any reason to that.

Some details

Lines 89 and 80. Please mention what the names stand for.

Line 60. That reference appears to give a study in the Antarctica. The quantitative photosynthesis by a sphagnum there may not be compared to that is a subtropical forest- or?

Line 115. Reomval should be removal.

Line 132 SOC has already been defined.

Line 143 what does ‘relative nmol per g dry soil mean’? Relative nmol?

Line 166 Fig 2G should be Fig 2F

Line 164 Delete ‘other’Review of the ms ‘Response of soil organic carbon stock to bryophyte removal……..’ by Chen et al. Ms ID Forests – 2010323

General comments

The authors have submitted a potentially interesting ms on the effects on the forest soil when bryophytes are removed. They used two forests; a subtropical one with evergreen broadleaf trees and a Sphagnum species as dominant bryophyte; the other forest was a temperate coniferous spruce and fir forest with a moss of the genus Actinothuidium.  Each investigation area was divided into six blocks each (6 control and 6 with the bryophytes removed). The experiment started in May 2017 and a sampling took place one year later. In the two forests the authors measured soil C and N as well as microbial biomass indicators. The authors conclude that the removal of these bryophytes has a clear influence on the sequestration of C and N in the forest floor. The study is interesting and could possibly be published after a rewriting. There is a number of problems with the present ms - please see below, and the English needs to be edited. I recommend a that the ms is rejected and that a resubmission is encouraged.

Specific comments

My major concern is the Methods’ section

The general design is very unclear. The authors removed the C- and N-containing bryophyte mass and measured the effects on the soil (apparently the mineral soil). Did the authors do this just once or did they repeat the removal? Did they remove the live parts of the plants or live parts plus attached dead – with e.g. Sphagnum this makes a clear difference. The authors investigated the soil after one year – as far as I can see they may have found a set of differences but I cannot find any explanation. In line 167 the authors claim that removal of bryophytes increased soil organic carbon stock (SOCs), which intuitively is contradictory as a source of C was removed. Still, I cannot find any explanation. Measurements of microbial biomass is interesting per se but does not explain any change in C and N.  Details below.  

In section 2.3 Soil measurements - the authors have not given any number for repetitions. They sampled for bulk density and I assume that they sampled the mineral soil; what depths? How many replicates. Do the authors refer to the 3 soil cores mentioned in line 121? These were sampled with one sample per plot. Is ‘plot’ the same as a block? This should be written more clearly.

Line 128 Soil water content was sampled – how many replicates? With what intensity - how often? Soil temperature - how many probes and for how long? How many replicates of soil pH? How many replicates of samples for soil C and N?  The analysis of C and N could be given in more detail? Did the authors need to separate organic and inorganic C? What digestion method was used?

Some more detail about the analysis for phospholipid fatty acids analysis is needed. The authors could also at least briefly mention what the Chao1 and Shannon indices are and what they tell the reader.

One piece of information is the duration of the experiment. As the authors give the effects of bryophyte removal on e.g. soil carbon and nitrogen – was this an experiment with just one removal of bryophyte and measurements made once a year after?

In the statistical analysis – how did the authors compare the effects at the two sites?

The Results and Discussion sections are in parts difficult to follow; some comments are given below.

Table 1. There are six columns giving F- and P- values. Some of the p values have a less-than–sign (<) others not and in some cases there should have been, I think, the sign =. Is that correct?

Figures 2 and 3. The significances given in the upper right corner of each subfigure – what do they say?

Figure 4 ‘Different lower-case letters mean……’  I cannot find any lower-case letters at all. Please add that or change the figure.  Subfigures C and D – what do they show? The legend needs to be developed.

Figure 5. Please define MSE.  I searched but could not find what the abbreviation stands for.

Lines 235. The sentence is a bit enigmatic. Please develop and explain so the reader can understand how the statement fits into the context. In fact, the whole first para (lines 233 thru 240) needs to be developed. The authors need to explain how litter decomposition rate has an effect.

Line 258. How does plant diversity come into this study? The authors did not study that and give no reference to their statement.

Line 262.  ‘nitrogen conversion’ ? Conversion to what?

The lines 261 to 265 appear enigmatic – the only thing they tell me is that there are different results reported but the authors do not suggest any reason to that.

Some details

Lines 89 and 80. Please mention what the names stand for.

Line 60. That reference appears to give a study in the Antarctica. The quantitative photosynthesis by a sphagnum there may not be compared to that is a subtropical forest- or?

Line 115. Reomval should be removal.

Line 132 SOC has already been defined.

Line 143 what does ‘relative nmol per g dry soil mean’? Relative nmol?

Line 166 Fig 2G should be Fig 2F

Line 164 Delete ‘other’Review of the ms ‘Response of soil organic carbon stock to bryophyte removal……..’ by Chen et al. Ms ID Forests – 2010323

General comments

The authors have submitted a potentially interesting ms on the effects on the forest soil when bryophytes are removed. They used two forests; a subtropical one with evergreen broadleaf trees and a Sphagnum species as dominant bryophyte; the other forest was a temperate coniferous spruce and fir forest with a moss of the genus Actinothuidium.  Each investigation area was divided into six blocks each (6 control and 6 with the bryophytes removed). The experiment started in May 2017 and a sampling took place one year later. In the two forests the authors measured soil C and N as well as microbial biomass indicators. The authors conclude that the removal of these bryophytes has a clear influence on the sequestration of C and N in the forest floor. The study is interesting and could possibly be published after a rewriting. There is a number of problems with the present ms - please see below, and the English needs to be edited. I recommend a that the ms is rejected and that a resubmission is encouraged.

Specific comments

My major concern is the Methods’ section

The general design is very unclear. The authors removed the C- and N-containing bryophyte mass and measured the effects on the soil (apparently the mineral soil). Did the authors do this just once or did they repeat the removal? Did they remove the live parts of the plants or live parts plus attached dead – with e.g. Sphagnum this makes a clear difference. The authors investigated the soil after one year – as far as I can see they may have found a set of differences but I cannot find any explanation. In line 167 the authors claim that removal of bryophytes increased soil organic carbon stock (SOCs), which intuitively is contradictory as a source of C was removed. Still, I cannot find any explanation. Measurements of microbial biomass is interesting per se but does not explain any change in C and N.  Details below.  

In section 2.3 Soil measurements - the authors have not given any number for repetitions. They sampled for bulk density and I assume that they sampled the mineral soil; what depths? How many replicates. Do the authors refer to the 3 soil cores mentioned in line 121? These were sampled with one sample per plot. Is ‘plot’ the same as a block? This should be written more clearly.

Line 128 Soil water content was sampled – how many replicates? With what intensity - how often? Soil temperature - how many probes and for how long? How many replicates of soil pH? How many replicates of samples for soil C and N?  The analysis of C and N could be given in more detail? Did the authors need to separate organic and inorganic C? What digestion method was used?

Some more detail about the analysis for phospholipid fatty acids analysis is needed. The authors could also at least briefly mention what the Chao1 and Shannon indices are and what they tell the reader.

One piece of information is the duration of the experiment. As the authors give the effects of bryophyte removal on e.g. soil carbon and nitrogen – was this an experiment with just one removal of bryophyte and measurements made once a year after?

In the statistical analysis – how did the authors compare the effects at the two sites?

The Results and Discussion sections are in parts difficult to follow; some comments are given below.

Table 1. There are six columns giving F- and P- values. Some of the p values have a less-than–sign (<) others not and in some cases there should have been, I think, the sign =. Is that correct?

Figures 2 and 3. The significances given in the upper right corner of each subfigure – what do they say?

Figure 4 ‘Different lower-case letters mean……’  I cannot find any lower-case letters at all. Please add that or change the figure.  Subfigures C and D – what do they show? The legend needs to be developed.

Figure 5. Please define MSE.  I searched but could not find what the abbreviation stands for.

Lines 235. The sentence is a bit enigmatic. Please develop and explain so the reader can understand how the statement fits into the context. In fact, the whole first para (lines 233 thru 240) needs to be developed. The authors need to explain how litter decomposition rate has an effect.

Line 258. How does plant diversity come into this study? The authors did not study that and give no reference to their statement.

Line 262.  ‘nitrogen conversion’ ? Conversion to what?

The lines 261 to 265 appear enigmatic – the only thing they tell me is that there are different results reported but the authors do not suggest any reason to that.

Some details

Lines 89 and 80. Please mention what the names stand for.

Line 60. That reference appears to give a study in the Antarctica. The quantitative photosynthesis by a sphagnum there may not be compared to that is a subtropical forest- or?

Line 115. Reomval should be removal.

Line 132 SOC has already been defined.

Line 143 what does ‘relative nmol per g dry soil mean’? Relative nmol?

Line 166 Fig 2G should be Fig 2F

Line 164 Delete ‘other’Review of the ms ‘Response of soil organic carbon stock to bryophyte removal……..’ by Chen et al. Ms ID Forests – 2010323

General comments

The authors have submitted a potentially interesting ms on the effects on the forest soil when bryophytes are removed. They used two forests; a subtropical one with evergreen broadleaf trees and a Sphagnum species as dominant bryophyte; the other forest was a temperate coniferous spruce and fir forest with a moss of the genus Actinothuidium.  Each investigation area was divided into six blocks each (6 control and 6 with the bryophytes removed). The experiment started in May 2017 and a sampling took place one year later. In the two forests the authors measured soil C and N as well as microbial biomass indicators. The authors conclude that the removal of these bryophytes has a clear influence on the sequestration of C and N in the forest floor. The study is interesting and could possibly be published after a rewriting. There is a number of problems with the present ms - please see below, and the English needs to be edited. I recommend a that the ms is rejected and that a resubmission is encouraged.

Specific comments

My major concern is the Methods’ section

The general design is very unclear. The authors removed the C- and N-containing bryophyte mass and measured the effects on the soil (apparently the mineral soil). Did the authors do this just once or did they repeat the removal? Did they remove the live parts of the plants or live parts plus attached dead – with e.g. Sphagnum this makes a clear difference. The authors investigated the soil after one year – as far as I can see they may have found a set of differences but I cannot find any explanation. In line 167 the authors claim that removal of bryophytes increased soil organic carbon stock (SOCs), which intuitively is contradictory as a source of C was removed. Still, I cannot find any explanation. Measurements of microbial biomass is interesting per se but does not explain any change in C and N.  Details below.  

In section 2.3 Soil measurements - the authors have not given any number for repetitions. They sampled for bulk density and I assume that they sampled the mineral soil; what depths? How many replicates. Do the authors refer to the 3 soil cores mentioned in line 121? These were sampled with one sample per plot. Is ‘plot’ the same as a block? This should be written more clearly.

Line 128 Soil water content was sampled – how many replicates? With what intensity - how often? Soil temperature - how many probes and for how long? How many replicates of soil pH? How many replicates of samples for soil C and N?  The analysis of C and N could be given in more detail? Did the authors need to separate organic and inorganic C? What digestion method was used?

Some more detail about the analysis for phospholipid fatty acids analysis is needed. The authors could also at least briefly mention what the Chao1 and Shannon indices are and what they tell the reader.

One piece of information is the duration of the experiment. As the authors give the effects of bryophyte removal on e.g. soil carbon and nitrogen – was this an experiment with just one removal of bryophyte and measurements made once a year after?

In the statistical analysis – how did the authors compare the effects at the two sites?

The Results and Discussion sections are in parts difficult to follow; some comments are given below.

Table 1. There are six columns giving F- and P- values. Some of the p values have a less-than–sign (<) others not and in some cases there should have been, I think, the sign =. Is that correct?

Figures 2 and 3. The significances given in the upper right corner of each subfigure – what do they say?

Figure 4 ‘Different lower-case letters mean……’  I cannot find any lower-case letters at all. Please add that or change the figure.  Subfigures C and D – what do they show? The legend needs to be developed.

Figure 5. Please define MSE.  I searched but could not find what the abbreviation stands for.

Lines 235. The sentence is a bit enigmatic. Please develop and explain so the reader can understand how the statement fits into the context. In fact, the whole first para (lines 233 thru 240) needs to be developed. The authors need to explain how litter decomposition rate has an effect.

Line 258. How does plant diversity come into this study? The authors did not study that and give no reference to their statement.

Line 262.  ‘nitrogen conversion’ ? Conversion to what?

The lines 261 to 265 appear enigmatic – the only thing they tell me is that there are different results reported but the authors do not suggest any reason to that.

Some details

Lines 89 and 80. Please mention what the names stand for.

Line 60. That reference appears to give a study in the Antarctica. The quantitative photosynthesis by a sphagnum there may not be compared to that is a subtropical forest- or?

Line 115. Reomval should be removal.

Line 132 SOC has already been defined.

Line 143 what does ‘relative nmol per g dry soil mean’? Relative nmol?

Line 166 Fig 2G should be Fig 2F

Line 164 Delete ‘other’

Reviewer 2 Report

The review of the manuscript titled "Response of soil organic carbon stock to bryophyte removal is regulated by forest type in Southwest China"

The study explores the effect of one-year bryophyte removal on soil organic carbon in evergreen broadleaved forest and temperate coniferous forest. This study has increasing importance in terms of the current decline in the population of bryophytes due to climate change. The authors provided interesting preliminary results after 1-year treatment, and I would suggest continuing the experiment to get a better understanding of the prolonged removal of bryophytes in the studied forest types. It would be interesting to expand the research to other latitudes (i.e., the permafrost area), and include additional parameters, such as soil microbial activity (BR and SIR). Though the manuscript is well written and logically structured, there are some small drawbacks (see the Specific Comments).

Specific comments:

L106. Correct "Abieg" to "Abies".

L116. Check "removal".

L185. Units are shown in Fig. 2G as "mg hm-2". Could it be mg/ha?

L210. What are the open red circles in Figs. 4 C and D? I suggest enlarging the signs on these plates; they are too small and the colors are similar.

L330. Check the references. There is a double numbering.

Round 2

Reviewer 1 Report

Review of the ms ‘Contrast responses of soil organic carbon stocks to bryophyte removal in two southwest forests of China’ by Chen et al. Ms ID Forests-2010323   Round 2

General comments

This ms I have reviewed before. The authors have improved the ms. However, reading it again I found some things that need to be commented on (below).

As far as I can see the authors need to make a limited number of corrections and I recommend minor revision.

Specific comments

In line 131 the authors mentionthe steel cylinder’ that they used for sampling but did not give details. How many replicate samples did the authors make for that sampling?

Line 153 ‘… a series of solvents…’ please give what the solvents were.

Line 122  there appears to be a missing word between blocks and twelve, alternatively  a  semicolon sign

Acknowledgements The authors thank Dr Anne-Lise Victoria. The name Victoria is a first name for women and very uncommon as a family name. It may be a good idea just to check that.
